# Effect of *Angelica gigas* Nakai Ethanol Extract and Decursin on Human Pancreatic Cancer Cells

**DOI:** 10.3390/molecules25092028

**Published:** 2020-04-27

**Authors:** Bitna Kweon, Yo-Han Han, Ji-Ye Kee, Jeong-Geon Mun, Hee Dong Jeon, Dae Hwan Yoon, Byung-Min Choi, Seung-Heon Hong

**Affiliations:** 1Department of Oriental Pharmacy, College of Pharmacy, Wonkwang-Oriental Medicines Research Institute, Wonkwang University, Iksan 54538, Korea; kbn306@naver.com (B.K.); dygks1867@hanmail.net (Y.-H.H.); keejy@wku.ac.kr (J.-Y.K.); wjdrjs05@daum.net (J.-G.M.); alen0707@naver.com (H.D.J.); ydh2715@gmail.com (D.H.Y.); 2Department of Herbology, School of Korean Medicine, Wonkwang University, Iksan 54538, Korea; 3Department of Biochemistry, School of Medicine, Wonkwang University, Iksan 54538, Korea

**Keywords:** *Angelica gigas* Nakai, decursin, ultra-performance liquid chromatography, cell cycle arrest, apoptosis, matrix metalloproteinase

## Abstract

Pancreatic cancer (PC) is one of the most severe cancers, and its incidence and mortality rates have steadily increased in the past decade. In this study, we demonstrate the effect of *Angelica gigas* Nakai extract on pancreatic ductal adenocarcinoma cells. We prepared *A. gigas* Nakai ethanol extract (AGE) using roots of *A. gigas* Nakai and detected its active compound decursin from AGE by ultra-performance liquid chromatography analysis. AGE and decursin significantly decreased viability and colony formation of PANC-1 and MIA PaCa-2 cells. AGE and decursin induced G0/G1 phase arrest through downregulation of cyclin D1 and cyclin-dependent kinase 4 (CDK4). Caspase-3-dependent apoptosis of PANC-1 cells was promoted by AGE and decursin. Additionally, nontoxic concentrations of AGE and decursin treatment could suppress matrix metalloproteinase (MMP)-2 and MMP-9 expression and activity by inhibiting p38 phosphorylation. Taken together, this study demonstrates that AGE and decursin have potential properties to be considered in PC treatment.

## 1. Introduction

According to GLOBOCAN 2018, a total of 458,918 new cases of pancreatic cancer (PC) and 432,242 new deaths were registered. In the last decade, the incidence and mortality rates of pancreatic cancer have increased worldwide [1]. Pancreatic cancer is one of the most lethal cancers because the five-year survival rate is less than 10%. The poor prognosis is due to difficulties in early detection and limited chemotherapeutic regimen [2].

The control of cell cycle and apoptosis is one of the important methods in cancer therapy as it can disrupt cancer cell proliferation and metastasis [3,4]. Cellular homeostasis is maintained by proliferation, differentiation, cell cycle progression, and apoptosis of cells [3]. Cell cycle checkpoints manage the order and accuracy of cell cycle progression [5]. Cyclin-dependent kinase complex (cyclin–CDK) is a protein complex that can regulate transcription, mRNA processing, and differentiation of cells. Once cyclin binds to CDK, this active-state complex can regulate cell cycle progression [6]. Apoptosis, or programmed cell death, occurs to remove defective cells by cellular degradation in multicellular organisms [3]. This self-destruction process can be widely triggered by several conditions, including extracellular stimuli, DNA breakdown, and deficiency of growth factor.

Matrix metalloproteinases (MMPs) are a family of zinc-containing enzymes that play important roles in cancer initiation, tumor growth, and metastasis in pathological conditions. MMP-2 and MMP-9 are gelatinases and can present proteolytic activity against extracellular matrix molecules, such as gelatin and type IV collagen. Several studies have revealed that MMP-2 and MMP-9 are correlated with poor prognosis in cancer patients because they are related to interaction of integrins for adhesion and invasion of cancer cells [7].

*Angelica gigas* Nakai is a medicinal herb in the Umbelliferae family. It has been applied to improve poor circulation, blood deficiency, and gynecologic diseases in East Asia. Decursin, which is one of the effective compounds of *Angelica gigas* Nakai, has diverse biological activities [8]. *Angelica gigas* Nakai and decursin have various pharmacological effects, such as anti-inflammatory, antiosteoclastic, and anticancer effects [9,10,11]. However, the inhibitory effect of *Angelica gigas* Nakai and decursin against pancreatic cancer has not been reported. Therefore, the main purpose of this study was to evaluate the inhibitory effect and related mechanisms of *Angelica gigas* Nakai ethanol extract (AGE) and decursin on pancreatic cancer cells.

## 2. Results

### 2.1. AGE and Decursin Inhibited Proliferation of PANC-1 and MIA PaCa-2 Cells

It has been reported that the extract of *Angelica gigas* Nakai contains decursin [8]. We first confirmed the presence of decursin in AGE by ultra-performance liquid chromatography (UPLC) assay conducted using AGE and decursin under the same conditions. The chemical structure of decursin is shown in Figure 1a. As shown in Figure 1b, decursin was detected in AGE at the same retention time as the standard decursin sample (Figure 1c). Subsequent experiments were carried out using AGE and decursin. To confirm whether AGE and decursin could selectively inhibit the viability of PC cells including PANC-1, MIA PaCa-2, and human pancreatic epithelial (HPNE) cells were treated with AGE and decursin for 72 h. As shown in Figure 2a, AGE decreased cell growth of PANC-1 and MIA PaCa-2 cells in a dose-dependent manner. A concentration of 100 µg/mL of AGE inhibited PANC-1 and MIA PaCa-2 cell viability up to approximately 30% and 73%, respectively. In addition, 60 µM of decursin decreased the viability of PC cells by 34% and 62%, respectively (Figure 2B). However, the viability of HPNE cells did not change by the same concentrations of AGE and decursin (Figure 2A,B). Additionally, colony formation of PANC-1 and MIA PaCa-2 cells was inhibited by AGE and decursin treatment (Figure 2C). Thus, up to 100 µg/mL of AGE and 60 µM of decursin can inhibit growth of PC cells except normal pancreatic cells.

### 2.2. AGE and Decursin Induced Cell Cycle Arrest of PC Cells by Decreasing Cyclin–CDK Expression

To confirm whether AGE and decursin can induce cell cycle arrest of PANC-1 cells, flow cytometry was conducted using AGE- and decursin-treated cells. The percentage of G0/G1 phase was dose-dependently increased by AGE and decursin in PANC-1 cells (Figure 3A,B). Previous studies have shown that a decrease in cyclin D1/CDK4 expression inhibits cell cycle progression in the G1 phase [4,12]. As shown in Figure 3C,D, protein and mRNA expression of cyclin D1 and CDK4 was dose-dependently decreased in PANC-1 cells by AGE and decursin treatment. These data indicate that AGE and decursin can induce cell cycle arrest of PC cells by reducing cyclin D1 and CDK4 expressions.

### 2.3. AGE and Decursin Induced Apoptosis of PC Cells

Annexin V and terminal deoxynucleotidyl transferase dUTP nick-end labeling (TUNEL) assay were performed to investigate whether AGE and decursin can decrease cell viability by inducing apoptosis of PC cells. AGE and decursin treatment significantly induced apoptotic cells by approximately 30% and 10%, respectively (Figure 4A,B). Moreover, TUNEL-positive cells were increased in AGE- and decursin-treated PANC-1 cells (Figure 4C,D). As expected, AGE and decursin promoted caspase-3 and poly(ADP-ribose) polymerase (PARP) cleavage in PANC-1 cells (Figure 4E,F). Therefore, AGE and decursin can decrease the viability of PC cells by inducing caspase-dependent apoptosis.

### 2.4. AGE and Decursin Decreased MMP-2 and MMP-9 Expression in PC Cells by Regulating p38 Phosphorylation

Several studies have revealed that decursin suppresses MMP-9 expression in CT26 colorectal cancer cells, MCF-7 breast cancer cells, and HT1080 fibrosarcoma cells [13,14,15]. It can also inhibit MMP-2 and MMP-9 activities in human umbilical vein endothelial cells (HUVEC) [16]. Therefore, we carried out an investigation to confirm whether AGE and decursin can inhibit MMP-2 and MMP-9 expression. As shown in Figure 5A,B, AGE and decursin decreased protein levels of MMP-2 and MMP-9 in both PC cells. Gelatinolytic activity of MMP-2 and MMP-9 was also suppressed by AGE and decursin treatment (Figure 5C,D). To determine which molecule is related to the inhibitory effect of AGE and decursin on MMP activity, phosphorylation of mitogen-activated protein kinases (MAPKs) was detected in PC cells. Among the MAPKs, p38 phosphorylation was commonly decreased in PANC-1 and MIA PaCa-2 cells (Figure 5E). These results suggested that MMP-2 and MMP-9 expression was controlled by suppressing phosphorylation of p38 in AGE- and decursin-treated PC cells.

## 3. Discussion

PC is one of the most serious cancers because of its late diagnosis and chemoresistance. The incidence rate of pancreatic cancer is also steadily increasing, and it ranks as the fourth leading cause of cancer mortality worldwide [17]. Although surgery and chemotherapy are common methods for PC treatment, more effective and safer therapeutic agents are required for PC treatment to increase general survival periods and reduce the side effects. Recent studies have reported that herbal products can sensitize PC cells to chemotherapy and target molecular factors that are involved in PC progression [18]. Therefore, we focused on the oriental herbal medicine *Angelica gigas* Nakai, which is most commonly used as traditional medicine in East Asia.

*Angelica gigas* Nakai contains coumarin constituents, such as decursin and decursinol angelate, and differ from *Angelica sinensis* and *Angelica acutiloba*. Decursin and decursinol angelate have been detected in the ethanol extract of *Angelica gigas* Nakai, and these compounds have an inhibitory effect on cancer progression. Notably, decursin inhibits various kinds of cancers, including lung cancer, melanoma, prostate cancer, bladder cancer, colon cancer, and lymphoma [11,19,20,21,22]. Thus, we first confirmed whether AGE contains decursin using UPLC assay. The results showed that decursin is the main constituent of AGE. According to a previous study, the second peak in the UPLC assay may indicate decursinol angelate [8]. In this study, AGE and decursin showed growth inhibition and colony formation of PANC-1 and MIA PaCa-2 cells in a dose-dependent manner.

Cell cycle arrest is known for regulating cancer cell proliferation by controlling the CDK–cyclin complex. Inactivation of CDK4–cyclin D complex can initiate G0/G1 arrest [4]. It has been reported that decursin, which is derived from *Angelica gigas* Nakai, can decrease cyclin D1 and CDK4 expression in 253J human bladder cancer cells and HCT116 human colon cancer cells [21]. In this study, AGE and decursin induced cell cycle arrest in the G0/G1 phase by downregulating cyclin D1 and CDK4 in PANC-1 and MIA PaCa-2 cells. Along with cell cycle arrest, apoptosis is a widely used pathway to treat cancers by eliminating cells that promote tumor initiation, progression, and metastasis [3,23,24]. Caspase-3 is involved in both intrinsic and extrinsic apoptotic pathway. Caspase-3 is fundamental in apoptotic chromatin condensation and DNA fragmentation [25,26,27,28]. PARP, which is catalyzed by caspase-3, is also indispensable in apoptosis as it reserves nicotinamide adenine dinucleotide (NAD) and ATP that are required for late apoptosis [29]. Extracts of *Angelica gigas* Nakai and decursin have been shown to induce apoptosis in several cancer cells, including lung cancer, melanoma, prostate cancer, and colon cancer, by increasing caspase-3 and PARP cleavage [11,19,20,21,22]. Our results demonstrate that AGE and decursin can promote apoptosis by inducing cleavage of caspase-3 and PARP in both PANC-1 and MIA PaCa-2 cells.

Activation of MMP-2 and MMP-9 is related to PC progression through invasion of cancer cells [30,31]. In addition, some studies have reported that natural products and their derived compounds exert antimetastatic effect by inhibiting the activity of MMP-2 and MMP-9 [32,33,34]. Among the multiple signaling pathways, the p38-dependent pathway is associated with expression and activation of MMPs in PC cells. As overexpression of p38 can increase MMP-9-dependent invasion of pancreatic cells, p38 is one of the target molecules to ameliorate PC [7]. Natural compounds, including silibinin and oxymatrine, have inhibitory effect on MMP-2 and MMP-9 via p38 inactivation in gastric and pancreatic cancer cells [35,36]. In our study, AGE and its effective compound decursin inhibited activation of MMPs by decreasing p38 phosphorylation in PANC-1 and MIA PaCa-2 cells in a dose-dependent manner. Therefore, the result indicates that AGE and decursin inhibit the invasion ability of PANC-1 and MIA PaCa-2 cells.

## 4. Materials and Methods

### 4.1. Reagents

Decursin (purity >95%) was purchased from Chengdu Biopurify Phytochemicals Ltd. (Chengdu, China). WST-1 reagent was purchased from Roche (Mannheim, Germany). Cyclin D1, CDK4, MMP-2, and MMP-9 antibodies were purchased from Santa Cruz Biotechnology (TX, USA). PARP, cleaved PARP, caspase-3, cleaved caspase-3, and α-tubulin antibodies were purchased from Cell Signaling Technology (Danvers, MA, USA). β-actin antibody was purchased from Thermo Scientific (Waltham, MA, USA). Antimouse and antirabbit secondary antibodies were purchased from Bethyl Laboratories (Montgomery, TX, USA) and Thermo Scientific, respectively.

### 4.2. Preparation of AGE

The roots of *Angelica gigas* Nakai were purchased from Human Herb (Gyeongsan, Korea). First, 70% ethanol (500 mL) was added to dried roots of *Angelica gigas* Nakai (50 g), and it was then boiled for 2 h. The extract was filtered, lyophilized, and dissolved in DMSO prior to in vitro experiment.

### 4.3. Ultra-Performance Liquid Chromatography

Detection of decursin from AGE was carried out using the 1290 Infinity UPLC System (Agilent Technologies, Santa Clara, CA, USA), and separation was done using Halo RP-amide 2.1 × 150 mm, 2 μm (AMT, Wilmington, DE, USA) diode array detector (DAD). The mobile phase was composed of 100% acetonitrile (A) and 0.1% H_3_PO_4_ in water (B) at a flow rate of 0.4 mL/min with 1 μL injection volume. The condition for analysis is presented in Table 1. The AGE sample and decursin were prepared by dissolving in methanol at a concentration of 5 and 1 mg/mL, respectively. After dissolving, samples were sonicated and filtered. Using a previously reported method [37], we calculated the quantity of decursin contained in AGE using peak area; 5 mg/mL of AGE is equal to 0.94 mg/mL of decursin.

### 4.4. Cell Lines and Cell Culture

PC cell lines PANC-1 and MIA PaCa-2 cells were obtained from American Type Culture Collection (ATCC, Manassas, VA, USA). Both cells were cultured in Dulbecco’s modified Eagle medium (DMEM) containing 10% fetal bovine serum (Hyclone, Logan, UT, USA) and 1% penicillin/streptomycin (Invitrogen Inc., Waltham, MA, USA) at 37 °C incubator in a humidified atmosphere containing 5% CO_2_. The HPNE was purchased from ATCC and cultured in 75% DMEM without glucose with additional 2 mM glutamine, 1.5 g/L sodium bicarbonate, and 25% Medium M3 base supplemented with 5% fetal bovine serum, 10 ng/mL human recombinant epidermal growth factor, 5.5 mM D-glucose, and 750 ng/mL puromycin at 37 °C incubator in a humidified atmosphere containing 5% CO_2_.

### 4.5. Measurement of Cell Viability

PANC-1, MIA PaCa-2, and HPNE cells (5 × 10^3^ cells/well) were incubated in a 96-well plate overnight. AGE (0, 25, 50, and 100 μg/mL) and decursin (0, 20, 40, and 60 μM) were added to cells and incubated for 72 h. WST-1 reagent was added to each well and incubated for 2 h. The absorbance was measured at 450 nm using a microplate reader.

### 4.6. Colony Formation

PANC-1 and MIA PaCa-2 cells (5 × 10^3^ cells/well) were seeded in a 12-well culture plate overnight. Cells were treated with various concentrations of AGE (0, 25, 50, and 100 μg/mL) and decursin (0, 20, 40, and 60 μM) for 15 days (PANC-1 cells) and 7 days (MIA PaCa-2 cells). Colonies were fixed with 4% formaldehyde and stained by 0.4% crystal violet.

### 4.7. Flow Cytometry Analysis for Cell Cycle Arrest

PANC-1 cells (2 × 10^5^ cells/well) were seeded into 6-well plates and treated with AGE (0, 25, 50, and 100 μg/mL) and decursin (0, 20, 40, and 60 μM) for 72 h. Cells were counted to adjust the concentration at 1 × 10^6^ cells/mL and washed twice with phosphate buffered saline (PBS). After fixation of cells with 70% ethanol overnight at −20 ℃, cells were centrifuged. The supernatant was removed, and pellets were washed with PBS. Muse cell cycle reagent was added to the pellets and incubated for 30 min at room temperature in the dark. Cell cycle distribution was analyzed by a Muse cell analyzer.

### 4.8. Terminal Deoxynucleotidyl Transferase dUTP Nick-End Labeling Assay

Cells (1.5 × 10^4^ cells/well) were seeded in a 8-well chamber slide and treated with various concentrations of AGE (0, 25, 50, and 100 μg/mL) and decursin (0, 20, 40, and 60 μM) for 72 h. Cells were fixed with 4% formaldehyde for 1 h and incubated on ice for 2 min with permeabilization solution. After washing with PBS, apoptotic cells were detected by the In Situ Cell Death Detection Kit, Fluorescein (Merck, Darmstadt, Germany). TUNEL-positive and DAPI-stained cells were photographed using the EVOS^®^ FL Cell Imaging System (Thermo Fisher Scientific, Waltham, MA, USA).

### 4.9. Annexin V Assay

Cells were treated with AGE (0, 25, 50, and 100 μg/mL) and decursin (0, 20, 40, and 60 μM) for 72 h. To adjust 1 × 10^5^ cells/mL in PBS, cells were centrifuged and counted. Then, 100 μL of cells were mixed with 100 μL of Muse annexin V and dead cell reagent, and cells were incubated for 20 min at room temperature in the dark. Annexin-positive cells were analyzed by a Muse cell analyzer (Millipore, Burlington, MA, USA).

### 4.10. Western Blotting

Cells (2 × 10^5^ cells/well) were seeded into a 6-well culture plate and stabilized overnight. Cells were treated with various concentrations of AGE (0, 25, 50, and 100 μg/mL) and decursin (0, 20, 40, and 60 μM) for 72 h. Radioimmunoprecipitation assay (RIPA) buffer (Thermo Scientific, MA, USA) was mixed with a protease inhibitor and phosphatase inhibitor to extract total proteins. Cell lysates were loaded on SDS-PAGE gel (8–12%) for electrophoresis and transferred to polyvinylidene difluoride membrane. After blocking with 5% bovine serum albumin (BSA) for 1 h at room temperature, membranes were incubated with primary antibodies overnight at 4 ℃. Following washing with Tris-buffered saline with 0.1% Tween 20 (TBST), the membranes were incubated with a suitable horseradish peroxidase (HRP)-conjugated secondary antibody for 1 h. The immunoreactivity bands were detected using WESTSAVE Femto (Ab Frontier, Seoul, Korea) and visualized by a FluoroChem E image analyzer (Cell Bioscience, Berkeley, CA, USA).

### 4.11. RNA Extraction and Real-Time RT-PCR

Total RNA was isolated from PANC-1 cells using a QIAzol lysis reagent (QIAGEN sciences, Germantown, MD, USA). The target cDNA was synthesized using a Power cDNA Synthesis Kit (Applied Biosystems, Carlsbad, CA, USA) and amplified using the primers indicated in Table 2. mRNA levels were determined with StepOnePlus real-time RT-PCR as previously described [12].

### 4.12. Detection of MMP-2 and MMP-9 Activity

MMP-2 and MMP-9 activity was detected by gelatin zymography. Gelatin zymography was conducted using a Zymogram Buffer Kit (Komabiotech, Seoul, South Korea) and the method described in a previous paper [12,34]. In brief, AGE and decursin were added to 5 × 10^5^ cells in 6-well plates and incubated for 24 h. Cell supernatant corresponding to equal volume of protein lysates was concentrated and mixed with 2× sample buffer, and electrophoresis was carried out. The gel was incubated in zymogram renaturing buffer and zymogram developing buffer. Activity of the MMPs was visualized by Coomassie blue R-250 solution.

### 4.13. Statistical Analysis

All data are expressed as the mean ± standard deviation (SD) of three independent experiments. Significant difference (* *p* < 0.05) was determined by the Student *t*-test.

## 5. Conclusions

This study demonstrates the inhibitory effects of AGE and decursin on viability and MMP activities of PC cells. AGE and decursin selectively inhibited proliferation of PANC-1 and MIA PaCa-2 cells by inducing cell cycle arrest and caspase-mediated apoptosis. In addition, AGE and decursin could suppress the expression and enzyme activity of MMP-2 and MMP-9 in PC cells. Thus, AGE and decursin are expected to be effective novel agents for inhibiting proliferation of PC cells.

## Figures and Tables

**Figure 1 molecules-25-02028-f001:**
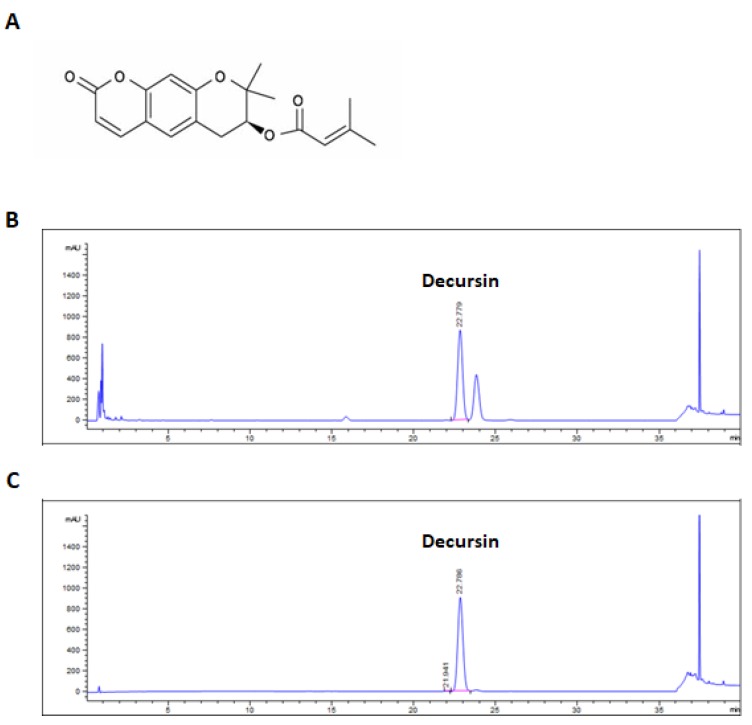
(**A**) Chemical structure of decursin. Ultra-performance liquid chromatography (UPLC) chromatograms of *Angelica gigas* Nakai ethanol extract (**B**) and the standard decursin (**C**).

**Figure 2 molecules-25-02028-f002:**
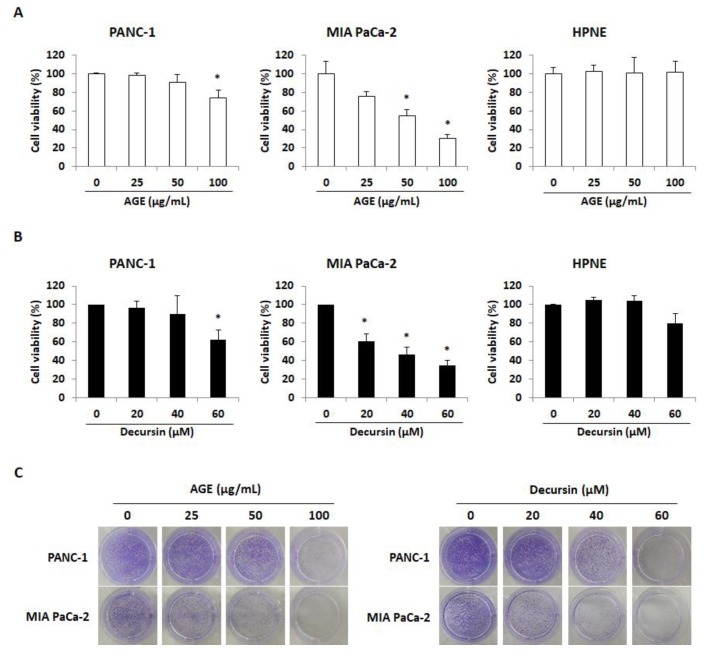
*Angelica gigas* Nakai ethanol extract (AGE) and decursin decreased viability and colony formation of pancreatic cancer (PC) cells. (**A**,**B**) Cell viability of PANC-1, MIA PaCa-2, and human pancreatic epithelial (HPNE) cells after AGE (**A**) and decursin (**B**) treatment for 72 h. (**C**) Colony formation results of AGE- and decursin-treated PANC-1 and MIA PaCa-2 cells. AGE (0–100 μg/mL) and decursin (0–60 μM) were used to treat PANC-1 and MIA PaCa-2 cells for seven days. Data were analyzed from at least three independent experiments. * *p* < 0.05.

**Figure 3 molecules-25-02028-f003:**
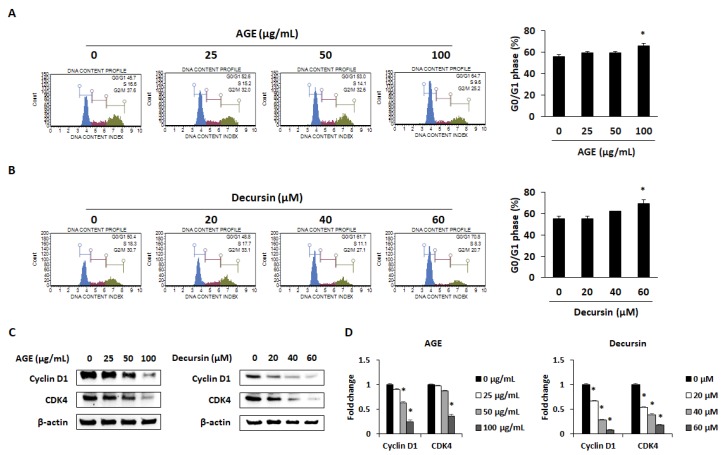
AGE and decursin induced G0/G1 phase cell cycle arrest of pancreatic cancer cells. (**A**,**B**) Cell cycle phase distribution of PANC-1 cells after AGE (**A**) and decursin (**B**) treatment was analyzed by flow cytometry. (**C**) Protein expression of cyclin D1 and CDK4 was detected in AGE- and decursin-treated pancreatic cancer cells. (**D**) mRNA expression of cyclin D1 and CDK4 was determined by real-time RT-PCR. Data were analyzed from at least three independent experiments. * *p* < 0.05.

**Figure 4 molecules-25-02028-f004:**
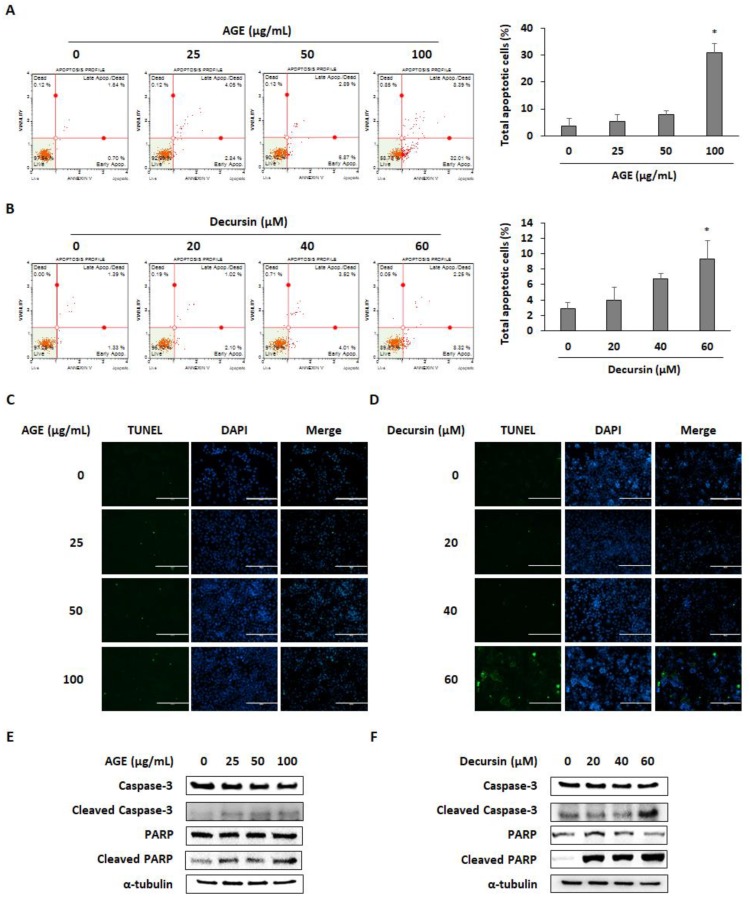
AGE and decursin induced apoptosis of pancreatic cancer cells. (**A**,**B**) The percentage of apoptotic cells after AGE (**A**) and decursin (**B**) treatment was detected by annexin V assay. (**C**,**D**) Terminal deoxynucleotidyl transferase dUTP nick-end labeling (TUNEL)-positive cells were captured by microscopic observation using AGE (**C**)- and decursin (**D**)-treated PANC-1 cells. (**E**,**F**) Protein expression of caspase-3, cleaved caspase-3, PARP, and cleaved PARP was detected in AGE (**E**)- and decursin (**F**)-treated PANC-1 cells. Data were analyzed from at least three independent experiments. * *p* < 0.05.

**Figure 5 molecules-25-02028-f005:**
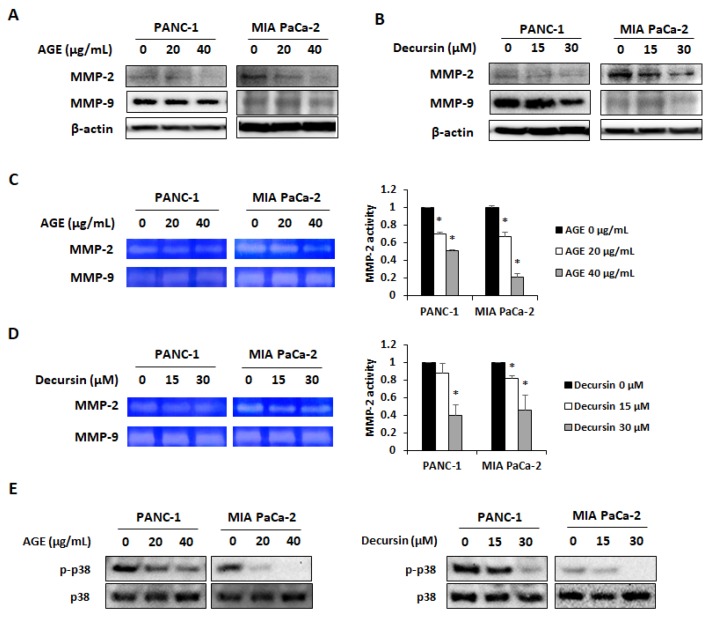
AGE and decursin suppressed expression and activity of MMP-2 and MMP-9 in PC cells. (**A**,**B**) Protein levels of MMP-2 and MMP-9 in PANC-1 and MIA PaCa-2 cells after AGE (**A**) and decursin (**B**) treatment. (**C**,**D**) Gelatin zymography results of AGE (**C**)- and decursin (**D**)-treated PC cells. (**E**) Changes of p38 phosphorylation by AGE and decursin in PC cells. Data were analyzed from at least three independent experiments. * *p* < 0.05.

**Table 1 molecules-25-02028-t001:** The condition of analysis of AGE by UPLC system.

Parameters	Condition		
Instrument	1290 Infinity UPLC System		
Detector	DAD		
Column	RP-Amide (2.1 × 150 mm, 2 μm)		
Mobile phase	A: 100% acetonitrile (*v*/*v*, %) B: 0.1% phosphoric acid in water (*v*/*v*, %)		
Gradient condition	Time (min)	A (%)	B (%)
	0	31	69
	35	31	69
	36	100	0
	40	100	0
Injection volume (μL)	0.1		
Flow rate (mL/min)	0.4		

**Table 2 molecules-25-02028-t002:** Sequences for real-time RT-PCR primers.

Genes	Forward (5′-3′)	Reverse (5′-3′)
Cyclin D1	ATGCCAACCTCCTCAACGAC	GGCTCTTTTTCACGGGCTCC
CDK4	GTGCAGTCGGTGGTACCTG	TTCGCTTGTGTGGGTTAAAA
GAPDH	CTGCACCACCAACTGCTTAG	TTCAGCTCAGGGATGACCTT

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
