# Peer review of "Effect of Angelica gigas Nakai Ethanol Extract and Decursin on Human Pancreatic Cancer Cells"

_molecules, 2020, doi:10.3390/molecules25092028_

Round 1

Reviewer 1 Report

Kweon et al. reported that Angelica gigas Nakai ethanol extract (AGE) and decursin can decrease pancreatic cancer cells viability and proliferation. The authors report that these are the first research results on the impact of AGE and decursin on pancreatic cell line. The results are interesting and show the mechanism of action.

Minor

Please correct the last sentence in the abstract and conclusion. We cannot directly draw such far-reaching conclusions based on in vitro studies on cell lines that these compounds have the potential for treatment of PC.

Author Response

Kweon et al. reported that Angelica gigas Nakai ethanol extract (AGE) and decursin can decrease pancreatic cancer cells viability and proliferation. The authors report that these are the first research results on the impact of AGE and decursin on pancreatic cell line. The results are interesting and show the mechanism of action.

Minor

Please correct the last sentence in the abstract and conclusion. We cannot directly draw such far-reaching conclusions based on in vitro studies on cell lines that these compounds have the potential for treatment of PC.

Answer: We appreciated your kindly review. As you comment, we edited the last sentence (Line 28, Line 296). To edit this sentence, we followed the other reviewer’s recommendation.

Reviewer 2 Report

The manuscript by Kweon et al. deals with the study of the effects of decursin, which is contained in Angelica gigas Nakai ethanol roots extract, on some pancreatic cancer cell lines. Although the beneficial effect of this molecule/extract in the treatment of several diseases is already known, the results reported in the manuscript extend the potentiality of the extracted molecule to pancreatic cancer (PC) treatment. However, the following items should be considered before the manuscript become acceptable for publication on Molecules.

Major comments

1) The results reported in the manuscript clearly indicate that the effect produced by decursin on PC cells can be observed on cell viability and colony formation through a G0/G1 cell cycle phase arrest, with an involvement of the caspase-3 dependent apoptosis, probably due to a suppression of the MMP-2/9 expression. Being these findings congruent with those recently reported for different natural extracts on other cancer cell lines (PMIDs 31817563, 28356942, 29450132, 15828817, 16431408, 24996346, etc. ) the Authors should consider the opportunity to cite these publications, in order to reinforce the general results that extract of natural origin can be used in cancer treatment, a statement that should be included i the discussion. In addition, an investigation on the effect of decursin on the cell migration/invasiveness of their cell lines should also be considered as this property is directly involved in cancer progression and metastatization.

2) In the paragraph 4.3, the indication of the methodology used for the quantification of "AGE samples" and "decursin" has to be described. Without this indication, the biological effect observed cannot be quantitatively evaluated.

3) In the paragraph 4.12, is the supernatant concentrated before using? Which is the amount of protein (in mass) that has been loaded onto the gel for the zymogram? How this amount has been normalized? Usually this normalization is carried out in comparison to the protein concentration in total extract of the colture (see PMIDs 31817563, 29450132, 25072751, etc.).

4) At least in the abstract (but also in the manuscript title", the indication that the extraction has been carried out on roots of A. gigas Nakai, should be stated.

Minor comments (general)

Use italics character when the name of Angelica gigas is displayed.

Minor comments (list)

Page 1

Line 17: delete "a".

Line 18: change "cascade" with "decade".

Line 19: add "extract" after "Nakai".

Line 26: change "to" with "properties to be considered in".

Line 35: change "are difficulty" with "is based on the difficulties"

Page 2

Line 45: change "at" with "in".

Line 49: change "gelatinase" with "gelatinases".

Line 53: and "a" before "medicinal".

Line 63: change "included" with "contained".

Line 64: delete the comma and "was"; add "by" after "AGE".

Page 3

Line 101: add "an investigation" after "out".

Line 103: change "Proteolytic" with "Gelatinolytic".

Page 7

Line 178: add "antibodies" after "MMP-9"

Line 179: add "antibodies" after "tubulin"

Line 180: add "antibody" after "actin"

Page 8

Line 199: change "fatal" with "fetal".

Line 204: delete "1 g/L".

Line 208: change "treated" with "added".

Line 218: change "adjusting" with "adjust".

Page 9

Line 262: either Table 2 or primer sequences have not been provided.

Author Response

Major comments

1) The results reported in the manuscript clearly indicate that the effect produced by decursin on PC cells can be observed on cell viability and colony formation through a G0/G1 cell cycle phase arrest, with an involvement of the caspase-3 dependent apoptosis, probably due to a suppression of the MMP-2/9 expression. Being these findings congruent with those recently reported for different natural extracts on other cancer cell lines (PMIDs 31817563, 28356942, 29450132, 15828817, 16431408, 24996346, etc. ) the Authors should consider the opportunity to cite these publications, in order to reinforce the general results that extract of natural origin can be used in cancer treatment, a statement that should be included i the discussion. In addition, an investigation on the effect of decursin on the cell migration/invasiveness of their cell lines should also be considered as this property is directly involved in cancer progression and metastatization.

Answer: As your comment, following reports (PMIDs 31817563 (Ref. 32), 28356942 (Ref. 33), 29450132 (Ref. 34), 15828817 (Ref. 27), 16431408 (Ref. 28), 24996346 (Ref. 24)) were described in discussion and reference section. Also, we indicated the anti-invasive effect of AGE and decursin in discussion section (Line 180-181).

2) In the paragraph 4.3, the indication of the methodology used for the quantification of "AGE samples" and "decursin" has to be described. Without this indication, the biological effect observed cannot be quantitatively evaluated.

Answer: We added in Materials and Methods section. Using previously reported method (PMID: 23924674), we calculated the quantity of decursin contained in AGE using peak area (5 mg/mL AGE ≒ 0.94 mg/mL decursin) (paragraph 4.3).

3) In the paragraph 4.12, is the supernatant concentrated before using? Which is the amount of protein (in mass) that has been loaded onto the gel for the zymogram? How this amount has been normalized? Usually this normalization is carried out in comparison to the protein concentration in total extract of the colture (see PMIDs 31817563, 29450132, 25072751, etc.).

Answer: Cell supernatant corresponding to equal volume of protein lysates was concentrated and was mixed with 2X sample buffer and carried out electrophoresis (the paragraph 4.12).

4) At least in the abstract (but also in the manuscript title", the indication that the extraction has been carried out on roots of A. gigas Nakai, should be stated.

Answer: As your comment, it has been described that A. gigas Nakai ethanol extract (AGE) was prepared using roots of A. gigas Nakai in the abstract.

Minor comments (general)

Use italics character when the name of Angelica gigas is displayed.

Minor comments (list)

Page 1

Line 17: delete "a".

Line 18: change "cascade" with "decade".

Line 19: add "extract" after "Nakai".

Line 26: change "to" with "properties to be considered in".

Line 35: change "are difficulty" with "is based on the difficulties"

Page 2

Line 45: change "at" with "in".

Line 49: change "gelatinase" with "gelatinases".

Line 53: and "a" before "medicinal".

Line 63: change "included" with "contained".

Line 64: delete the comma and "was"; add "by" after "AGE".

Page 3

Line 101: add "an investigation" after "out".

Line 103: change "Proteolytic" with "Gelatinolytic".

Page 7

Line 178: add "antibodies" after "MMP-9"

Line 179: add "antibodies" after "tubulin"

Line 180: add "antibody" after "actin"

Page 8

Line 199: change "fatal" with "fetal".

Line 204: delete "1 g/L".

Line 208: change "treated" with "added".

Line 218: change "adjusting" with "adjust".

Page 9

Line 262: either Table 2 or primer sequences have not been provided.

Thank you for your kindly review. We edited all of parts and added Table 2 as you recommended.
